# Acetate-Mediated Odorant Receptor OR51E2 Activation Results in Calcitonin Secretion in Parafollicular C-Cells: A Novel Diagnostic Target of Human Medullary Thyroid Cancer

**DOI:** 10.3390/biomedicines11061688

**Published:** 2023-06-11

**Authors:** Hyeon Jeong Lee, Cheol Ryong Ku, Arthur Cho, TaeHo Cho, ChaeEun Lee, Chan Woo Kang, Daham Kim, Yoon Hee Cho, JaeHyung Koo, Eun Jig Lee

**Affiliations:** 1Endocrinology, Institute of Endocrine Research, Department of Internal Medicine, Yonsei University College of Medicine, Seoul 03722, Republic of Korea; 2Department of Nuclear Medicine, Yonsei University College of Medicine, Seoul 03722, Republic of Korea; 3Department of New Biology, DGIST, Daegu 42988, Republic of Korea; 4Brain Korea 21 Project for Medical Science, Yonsei University College of Medicine, Seoul 03722, Republic of Korea

**Keywords:** odorant receptors, acetate, calcitonin, parafollicular C-cells, medullary thyroid cancer, positron emission tomography

## Abstract

Medullary thyroid cancer originates from parafollicular C-cells in the thyroid. Despite successful thyroidectomy, localizing remnant cancer cells in patients with elevated calcitonin and carcinoembryonic antigen levels remains a challenge. Extranasal odorant receptors are expressed in cells from non-olfactory tissues, including C-cells. This study evaluates the odorant receptor signals from parafollicular C-cells, specifically, the presence of olfactory marker protein, and further assesses the ability of the protein in localizing and treating medullary thyroid cancer. We used immunohistochemistry, immunofluorescent staining, Western blot, RNA sequencing, and real time-PCR to analyze the expression of odorant receptors in mice thyroids, thyroid cancer cell lines, and patient specimens. We used in vivo assays to analyze acetate binding, calcitonin secretion, and cAMP pathway. We also used positron emission tomography (PET) to assess C^11^-acetate uptake in medullary thyroid cancer patients. We investigated olfactory marker protein expression in C-cells in patients and found that it co-localizes with calcitonin in C-cells from both normal and cancer cell lines. Specifically, we found that OR51E2 and OR51E1 were expressed in thyroid cancer cell lines and human medullary thyroid cancer cells. Furthermore, we found that in the C-cells, the binding of acetate to OR51E2 activates its migration into the nucleus, subsequently resulting in calcitonin secretion via the cAMP pathway. Finally, we found that C^11^-acetate, a positron emission tomography radiotracer analog for acetate, binds competitively to OR51E2. We confirmed C^11^-acetate uptake in cancer cells and in human patients using PET. We demonstrated that acetate binds to OR51E2 in C-cells. Using C^11^-acetate PET, we identified recurrence sites in post-operative medullary thyroid cancer patients. Therefore, OR51E2 may be a novel diagnostic and therapeutic target for medullary thyroid cancer.

## 1. Introduction

Medullary thyroid cancer (MTC), a neuroendocrine tumor originating in the parafollicular cells (C-cells) [1] in the thyroid, is associated with high rates of recurrence and metastasis [2]. MTC recurrence is clinically suspected when serum calcitonin levels increase after thyroidectomy. High-risk factors for thyroid cancer include genetic factors, radiation exposure, and thyroid dysfunction, as well as a calcitonin serum concentration of 100 pg/mL or higher [3]. However, it is challenging to localize the sites of remnant cancer cells or potential tumor recurrence after thyroidectomy. Various modalities, such as octreotide scans, 18-Fluore-dihydroxyphenylalanine (F-18 DOPA), or fluorodeoxyglucose positron emission tomography (PET), have been utilized to identify recurrence sites; however, their efficacy remains uncertain [4,5,6].

Odorant receptors (ORs) are the largest subfamily of G protein-coupled receptors (GPCRs). They function as sensors of small metabolites and have recently been found in non-olfactory tissues [4,5]. Extranasal ORs, such as olfactory marker protein (OMP), olfactory G-protein, and adenylate cyclase III, are expressed in non-olfactory tissues, including the kidneys, skin, and C-cells of the thyroid [6]. As a prominent example of expression in prostate cancer, upregulation of OR51E1 and OR51E2 is observed in benign prostatic tissue. It is known that their overexpression inhibits cell proliferation and promotes cell death. Additionally, OR51E2 inhibits the growth of prostate cancer but promotes invasiveness and metastasis [7,8]. However, no report to date has explored the potential application of extranasal OR for diagnosis or its “druggability” in human diseases. There have been a number of recent proposals for targeting ORs with therapeutic drugs [9,10]. Therefore, we hypothesized that C-cell OR signals, particularly OMP, could be clinically useful in localizing and treating MTC.

## 2. Materials and Methods

### 2.1. Cell Culture

TT medullary thyroid cancer and FRO, SW1736 anaplastic thyroid cell lines were grown in RPMI 1640 medium containing 10% fetal bovine serum (FBS), 100 units/mL penicillin, and 100 µg/mL streptomycin. The MZ-CRC-1 MTC, TPC-1 papillary thyroid cancer, and FTC-133 follicular thyroid cancer cell lines were cultured in DMEM containing 10% FBS and 100 units/mL penicillin and 100 µg/mL streptomycin in a 5% CO_2_ incubator at 37 °C.

TT cells were purchased from American Type Culture Collection and FRO. TPC-1, FTC-133, and MZ-CRC-1 cells were provided by Professor Woong Youn Chung (Yonsei medical center, Seoul, Republic of Korea). SW1736 cells were provided by Professor Yoon Woo Koh (Yonsei medical center, Seoul, Republic of Korea). All cell culture reagents were supplied by Hyclone Laboratories, Inc. (Logan, UT, USA).

### 2.2. RNA Isolation, RNA Sequencing, and Quantitative Real-Time PCR (qRT-PCR)

Total RNA was isolated using NucleoZOL Reagent (MN, Gaithersburg, MD, USA) as per manufacturer’s instruction. For RNA sequencing (RNA-seq), sequencing library was prepared using NEBNext RNA prep kit (NEB, Ipswich, MA, USA). Sequencing was performed using HiSeq 2500 (Illumina, San Diego, CA, USA) with rapid run 150 bp PE mode. Differential expression of OR was confirmed using RPKM (reads per kilo base per million). qRT-PCR analysis was performed according to standard procedure using a ReverTra Ace-α First Strand cDNA Synthesis kit (Toyobo, Osaka, Japan).

qRT-PCR was performed using primers specific for *OR51E1* (sense: 5′-TACATTGTGCGGACTGAGCA-3′, antisense: 5′-CCAACTAGCGGTCAAAAGCC-3′), *OR51E2* (sense: 5′-TGCATCGTGGTCTTCATCGT-3′, antisense: 5′-TCTGGGTAAGACAGGCCTCA-3′), *OMP* (sense: 5′-TGTGTACCGCCTCAACTTCA-3′, antisense: 5′-GTCGGCCTCATTCCAATCTA-3′), *calcitonin* (sense: 5′-CCAGGTGCTCCAACCCC-3′, antisense: 5′-GGCAGCCTCCATGCAGCAC-3′), and *GAPDH* (sense: 5′-CAAGGTCATCCATGACAACT-3′, antisense: 5′-TTCACCACCTTCTTGATGTC-3′). The cycling conditions were denaturation at 95 °C for 5 min, followed by 35 cycles of 95 °C for 45 s, 60 °C for 45 s, and 72 °C for 45 s. All reactions were performed in triplicate. Relative mRNA expression was analyzed using the 2^−ΔΔCt^ method, using *GAPDH* expression for normalization.

### 2.3. Luciferase Assay

The Dual-Glo luciferase assay system (#E2940; Promega, Madison, WI, USA) was used for OR ligand–receptor interaction as previously described [11]. Briefly, 200 μL of MZ-CRC-1 cells (1 × 10^4^ cells/well) was seeded in white polystyrene 96-well plates (#353296; BD Biosciences, San Jose, CA, USA) 1 day before transfection. In each well, OR51E2, CRE-luciferase reporter, and Renilla expression vectors were transiently co-transfected into the cells using Lipofectamine 2000. Transfected cells were stimulated at 37 °C for 4 h with various concentrations of acetate diluted in CD293 medium (#11913-019; Thermo Fisher Scientific, Waltham, MA, USA). The activity of firefly luciferase was normalized against that of Renilla luciferase. Luminescence was measured on a SpectraMax L microplate reader (Molecular Devices, San Jose, CA, USA).

### 2.4. cAMP Assay

cAMP assay was performed according to manufacturer’s instruction (Cyclic AMP XP assay kit, #4339; Cell Signaling Technology, Beverly, MA, USA). Briefly, 5 mL of cells (2 × 10^6^ cells/well) was plated on 60 mm dish, and after overnight adhesion, treated with vehicle or 250 µM acetate and incubated for 1 h. Cells were rinsed with PBS and iced for 10 min after adding 100 μL of 1× lysis buffer. A standard was prepared in 1× lysis buffer and 50 uL HRP-linked cAMP solution, and 50 uL sample to the cAMP plate was added. After incubation at room temperature (RT) for 3 h on a horizontal orbital plate shaker, plate contents were discarded and washed 4 times with 1× wash buffer. The buffer in the well was discarded and dried completely. A total of 100 μL TMB substrate was added and incubated at RT for 30 min. A total of 100 μL stop solution was added, and absorbance at 450 nm was measured.

### 2.5. Small Interfering RNA (siRNA) Transfection

OR51E2 was knocked down using Ambion^®^ Silencer^®^ select pre-designed siRNA (siRNA ID: s37520, Life Technologies, Carlsbad, CA, USA). A non-specific control pool was used for the stealth RNAi negative control (12935-100; Invitrogen, Waltham, MA, USA).

MZ-CRC-1 cells were transfected with siRNA using Lipofectamine 2000 (Invitrogen, Carlsbad, CA, USA) as per manufacturer’s instructions. Briefly, 200 nM of siRNA and Lipofectamine 2000 was added separately to OPTI MEM medium (Gibco, Grand Island, NY, USA). After 5 min, the two solutions were mixed and incubated for 20 min at RT. The mixture was added to monolayer of cells seeded in 60 mm culture dishes. Medium was replaced with complete medium 6 h after transfection, and siRNA-mediated knockdown experiment was performed after 48 h or 72 h.

### 2.6. Calcitonin Secretion Assay

For in vitro studies, MZ-CRC-1 cells were seeded in a 6-well plate in a volume of 2 mL cell suspension medium so that 1 × 10^6^ cells were entered into each well and cultured overnight. The medium was replaced with serum-free medium the next day, followed by treatment with 250 μM acetate (Sigma-Aldrich, St. Louis, MO, USA). Culture medium was collected at 0, 1, 3, 5, and 10 min after acetate administration, and calcitonin levels were measured using a commercial ELISA Kit for Calcitonin (CEA472Hu; Cloud-Clone Corporation, Houston, TX, USA), as per manufacturer’s instructions.

### 2.7. Animal Studies

All animal experiments were approved by the Institutional Animal Care and Use Committee at the Yonsei University Health System. C57/BL6 mice were obtained from Central Lab. Animal Inc. (Seoul, Republic of Korea).

*Olfr78*-deficient mice (*Olfr78*-KO) were a gift from Dr. Jennifer Pluznick (Johns Hopkins University School of Medicine, Baltimore, MD, USA) and were created by targeting the *Olfr78* gene and replacing it with GFP and tau-lacZ. For our experiments, heterozygous females were mated to *Olfr78*-KO males to produce litters with males whose genotype was either wild-type (WT) or *Olfr78*-KO. Genotypes were determined via PCR analysis of tail sample DNA. WT mice are littermates of *Olfr78*-KO mice. The primer information used for genotyping is as follows: Common_R: 5′-GCATACATGATACACATAAGCCTTC-3′, WT_F1: 5′-CACTCATTGGTCTGTCAGT GG-3′ Mutant_F2: 5′-CTACCATTACCAGTTGGTCTGGTG-3′. In the PCR results, if the product size was 630 bp, it was labeled as WT; if 683 bp was found, it was labeled as KO; and if both bands were present, it was labeled as HT.

Eight-week-old KO and WT mice were used for serial serum calcitonin measurements after acetate stimulation. Retro-orbital blood collection was performed immediately before and 30 min after acetate administration (500 mg/kg in normal saline, all mice received 100 μL i.p. injection, each *n* = 6).

Serum samples were obtained after incubation for 20 min at room temperature and centrifugation at 2000 rpm for 15 min at 4 °C. Mice serum calcitonin levels were measured using a commercial ELISA Kit for Calcitonin (CEA472Mu; Cloud-Clone Corporation, Houston, TX, USA) according to the manufacturer’s instructions.

### 2.8. Immunofluorescence

MZ-CRC-1 and TT cells (2 × 10^4^ cells/well) were grown on poly-D-lysine coated confocal dish and allowed to adhere overnight. Cells were washed with PBS and fixed for 10 min with 4% paraformaldehyde (DAEJUNG, Seoul, Republic of Korea). Cells were permeabilized with 0.1% Triton X-100 (Promega Corporation, Madison, WI, USA) for 5 min and blocked in 5% donkey serum (Vector Laboratories, Burlingame, CA, USA). Cells were incubated with anti-OR51E2 (Novus Biologicals, Centennial, CO, USA) and OMP (Wako Pure Chemical Industries, Osaka, Japan) primary antibody at 1:500 in blocking buffer, followed by donkey anti-rabbit-FITC or anti-mouse Cy3 secondary antibody (Jackson Immuno Research, West Grove, PA, USA) at 1:50. Antibodies were incubated for 1 h at RT. OR51E2 and Olfr78 are same orthologues for olfactory receptors, so OR51E2 was used for human tissues, and Olfr78 was used for rodent tissues. All cells were also counterstained with DAPI mounting solution. Images were captured using a confocal laser scanning microscope LSM700 and processed using ZEN software 2009 light edition (Carl Zeiss, Oberkochen, Germany).

C57/BL6 WT mice and KO mice were anesthetized with 7 mg/kg Zoletil (Virbac, SA, Carros Cedex, France) and perfused and fixed trans-cardially with PBS (pH 7.4) and ice-cold freshly prepared 4% PFA in PBS using a peristaltic pump (AC-2110, ATTO, Tokyo, Japan). Buffer speed during perfusion was maintained at 3 mL/min with a peristaltic pump. Thyroid tissues were isolated and post-fixed for 2 h in cold 4% PFA (DAEJUNG, Seoul, Republic of Korea). After cryoprotection in 30% sucrose in PBS, thyroid tissues were embedded in an OCT compound (Tissue Tek, Sakura Finetek, Torrance, CA, USA) and snap-frozen on dry ice. The tissues were cryosectioned (14 μm), placed on micro slides (5116-20F, Muto Pure Chemicals, Tokyo, Japan), and dried at 37 °C for 30 min. The tissues were then incubated for 3 days at 4 °C with goat anti-OMP (1:2500, Wako Pure Chemical Industries, Osaka, Japan), rabbit anti-calcitonin (1:10,000, DakoCytomation, Glostrup, Denmark), and rabbit anti-OR51E2 (Olfr78 human ortholog, 1:200, LifeSpan Biosciences, Seattle, WA, USA), all diluted in PBS with 0.3% Triton X-100 and 10% normal horse serum (NHS; Vector Laboratories, Burlingame, CA, USA). After three washes in PBS, the bound primary antibodies were detected via incubation with Cy3- or Alexa Fluor 488-conjugated secondary antibodies diluted in PBS containing 0.3% Triton X-100 and 10% NHS for 2 h in the dark. The slides were treated with DAPI (Vector Laboratories, Burlingame, CA, USA) before visualization. Samples were visualized with an Axioskop microscope (Carl Zeiss, Oberkochen, Germany).

### 2.9. Protein Extraction and Western Blotting

MZ-CRC-1 cell lysates were prepared with lysis buffer (50 mM Tris-HCl, 150 mM sodium chloride, 1% Triton X-100, 1% sodium deoxycholate, 0.1% SDS, pH 7.5, and 2 mM EDTA, 1 mM sodium orthovanadate, 0.1 mM phenylmethylsulfonyl fluoride (PMSF), 0.5% NP-40, protease inhibitor cocktail (Sigma-Aldrich, St. Louis, MO, USA)). The protein concentration was determined using the Bradford assay (Bradford assay kit, Biorad Laboratories, CA, USA) with bovine serum albumin (BSA) as the standard. Equal aliquots of total cell lysates (30 μg) were solubilized in sample buffer and electrophoresed on denaturing SDS-polyacrylamide gel (10% and 12% separating gel). The proteins were transferred to polyvinylidene difluoride membranes. The membranes were blocked with 5% nonfat dry milk in TBS containing 0.05% Tween 20 for 1 h at RT and incubated with primary antibodies overnight at 4 °C and then with horseradish peroxidase-conjugated secondary for 2 h at RT. Primary antibodies used CREB (#9197; Cell signaling, Danvers, MA, USA), phospho-CREB (ser133) (#9198; Cell signaling, Danvers, MA, USA), and OR51E2 (NBP1-71145; Novus Biologicals, Centennial, CO, USA). β-Actin (C4) HRP (47778; Santa Cruz Biotechnology, Santa Cruz, CA, USA) was used as a loading control. The secondary antibody used was goat anti-rabbit immunoglobulin-HRP antibody (sc-2004; Santa Cruz Biotechnology). β -Actin was diluted 1:3000 in 5% nonfat dry milk. CREB, phospho-CREB, and OR51E2 were added 1:1000 in 5 % BSA. The secondary antibody was added 1:3000 in 5% nonfat dry milk. Antigen-antibody complexes were detected with WEST-SAVE Up™ luminol-based ECL reagent (ABfrontier, Seoul, Republic of Korea).

Protein was extracted from MTC patient specimens. Western blotting was performed using established procedures with modifications. Briefly, samples were homogenized in 100 μL RIPA buffer (25 mM Tris-HCl pH 7.6, 150 mM NaCl, 1% NP-40, 1% sodium deox-ycholate, 0.1% SDS with 0.1 mM PMSF and 1× protease inhibitors) using a TissueLyser II (Qiagen, Hilden, Germany). A total of 10 μg of each sample was loaded onto an SDS gel and transferred by electrophoresis to PVDF membrane. Membranes were blocked with 5% nonfat dry milk in TBS containing 0.05% Tween 20 for 1 h at RT. OR51E2 primary antibody (NBP1-71145; Novus Biologicals, Centennial, CO, USA) was diluted 1:500 in 5% BSA buffer and incubated overnight. Blots were washed three times with 0.05% TBS-T buffer and then incubated with HRP-conjugated secondary antibody for 1 h at RT. As the secondary antibody, goat anti-rabbit IgG-HRP antibody (sc-2004; Santa Cruz Biotechnology) was diluted 1:3000 in 5% nonfat dry milk. Western blot results were normalized with β-Actin (1:3000, 47778; Santa Cruz Biotechnology). Beta-Actin was diluted 1:3000 in 5% nonfat dry milk and reacted for 1 h at RT. Immunoreactivity was detected with WEST-SAVE UpTM luminol-based ECL reagent (ABfrontier).

### 2.10. In Vitro and In Vivo C^11^-Acetate Uptake Assays

Thyroid cancer cells were plated in 6 well plates (1 × 10^6^ cells/well), and in vitro C^11^-acetate uptake assays were performed at 85–95% confluency. C^11^-acetate treatment was performed for 10 min (at 2, 5, 10, 20, 40, or 60 µCi). The cells were washed thrice with PBS, lysed with RIPA buffer, scraped and collected, and counted using a Wallac Wizard 1480 gamma counter (PerkinElmer Inc., Akron, OH, USA). Protein concentration was determined using a Bradford assay kit (Bio-Rad, Hercules, CA, USA).

For C^11^-acetate competitive binding assays (saturation kinetic assay), non-radioactive acetate treatment was performed for 3 min (at 0, 20, 50, 80, 150, and 300 µM). After PBS wash, 10 µCi C^11^-acetate was added to each well, followed by incubation for 10 min. The samples were washed with PBS, lysed, and gamma counted. Gamma counts were normalized to protein concentration.

For C^11^-acetate uptake assays, *OR51E2* siRNA knockdown MZ-CRC-1 cells were treated with 10 µCi C^11^-acetate for 10 min and evaluated as described above.

In vivo C^11^-acetate uptake studies were performed in *Olfr78*-KO and WT mice. Each mouse was fed food and water ad libitum, and 300 ± 20 µCi was injected via the tail vein. Mice were sacrificed 10 min after injection. Approximately 0.1 mL of blood was obtained through retro-orbital sampling using a heparinized capillary tube, and the blood was weighed and gamma counted for injected dose normalization. The thyroid gland of each mouse was also weighed, gamma counted, and normalized for injected dose by dividing it by blood radioactivity.

### 2.11. C^11^-Acetate PET/Computed Tomography (PET/CT)

A patient underwent pre-operative C^11^-acetate PET/CT scanning with a hybrid PET/CT scanner (Discovery 710, GE Healthcare, Milwaukee, WI, USA). C^11^-acetate was injected (5.5 MBq/kg), and C^11^-acetate PET and low-dose CT images (30 mA and 103 kVp) were obtained 20 min after C^11^-acetate injection. PET acquisition time was 3 min/bed position in the three-dimensional mode, and images were reconstructed in ordered subset expectation maximization with attenuation correction. An additional spot view of the neck area was acquired immediately after whole-body acquisition at the 5 min/bed position.

This study was conducted in accordance with the Declaration of Helsinki and approved by our institutional review board (No. 1-2016-0033).

### 2.12. Statistical Analysis

Results are expressed as mean ± standard error of mean. Statistical analysis was performed using GraphPad Prism software v.4.0.0 (GraphPad, Inc., La Jolla, CA, USA). Statistical significance was determined using one-way ANOVA, followed by Student’s *t*-test and Mann–Whitney test to compare the means of two different groups. *p*-values < 0.05 were considered statistically significant.

## 3. Results

Immunohistochemical analyses revealed co-localization of OMP with calcitonin in C-cells from both normal thyroid glands and MTC (Figure 1A), which is consistent with the findings of a previous study [6]. Several ORs containing *Olfr78* (human ortholog *OR51E2*) and *Olfr558* (human ortholog *OR51E1*) are expressed with OMP in mouse thyroids [6]. RNA sequencing and qRT-PCR analysis of *OR51E2* and *OR51E1* expression were performed in medullary (MZ-CRC-1 and TT), follicular (FTC-133), papillary (TPC-1), and anaplastic (FRO) thyroid cancer cell lines (Figure 1B and Appendix A). Compared to other cell lines, *OR51E2* was highly (*p* < 0.01) expressed with *OR51E1* in MTC cell lines. Additionally, the expression of OR51E2 with OMP was confirmed in the MZ-CRC-1 and TT cell lines using immunofluorescent staining (Figure 1C). In a pathological specimen from a 31-year-old male patient with MTC, hematoxylin and eosin, immunohistochemistry, and Western blot analysis revealed a normal thyroid gland with low OR51E2 expression and high OR51E2 expression in the primary MTC (Figure 1D,E).

We then evaluated whether acetate-stimulated OR51E2 increases calcitonin secretion through the cAMP pathway since small-chain fatty acids, such as acetate and propionate, have been reported as OR51E2 ligands [12]. We demonstrated acetate-OR51E2 reactivity in a concentration-dependent manner based on luciferase activity (Figure 2A). The acetate-activated cAMP pathway increased cAMP and phosphorylated cAMP response element-binding protein (CREB) in the MZ-CRC-1 cells (Figure 2B,C). Additionally, acetate treatment increased calcitonin secretion significantly in a time-dependent manner, whereas *OR51E2* siRNA knockdown eliminated the effect (Figure 2D).

We questioned whether we would be able to see a recapitulation of the effect of acetate on calcitonin secretion in an animal model. Olfr78, the mouse ortholog of OR51E2, has been studied in an animal model, the *Olfr78* knockout (*Olfr78*-KO) mouse [12]. We observed the co-localization of Olfr78 with OMP and the co-localization of calcitonin and Olfr78-GFP replacement using immunofluorescence, validating the suitability of the animal model for further analysis (Figure 3A). Subsequently, we compared changes in serum calcitonin levels, before and after acetate administration, in WT and *Olfr78*-KO mice. After acetate administration, calcitonin levels increased by 1.35 ± 0.15 on average in WT mice (*p* = 0.041); however, there was no significant change in calcitonin levels in *Olfr78*-KO mice (1.068 ± 0.059, *p* = 0.273, Figure 3B).

We further evaluated whether acetate is internalized along with OR51E2 in MTC. Some GPCRs, such as CCR2, IGF-1R, CXCR4, and oxytocin receptors, have been characterized well by signal transduction cascades in the plasma membrane as well as receptor internalization and nuclear translocation [13,14,15,16]. We investigated whether OR51E2 migrates into the nucleus through acetate-activated nuclear trafficking in MZ-CRC-1 cells using immunofluorescence techniques. Migration of OR51E2 to the nucleus was observed within 1 min of acetate treatment and persisted for up to 30 min (Figure 4A). We have shown that systematic acetate administration increases serum calcitonin, presumably via OR51E2 activation in parafollicular cell lines. The findings suggest that a radioactive acetate analog, such as C^11^-acetate, may be a useful PET imaging marker for OR51E2. To test this hypothesis, we performed in vitro C^11^-acetate uptake studies using various thyroid cancer cell lines. Consistent with previous results, we observed a 100–600% higher uptake in MTC than in non-MTC cell lines (Figure 4B). C^11^-acetate titration studies revealed approximate IC_50_ values of 30 and 20 µM for MZ-CRC-1 and TT MTC cell lines, respectively. Competitive studies using non-radioactive acetate revealed similar IC_50_ values, suggesting that C^11^-acetate and acetate have similar binding affinities for OR51E2. EC_50_ values were 138.1 µCi for TT and 59.6 µCi for MZ-CRC-1. IC_50_ values for TT and MZ-CRC-1 were 16.86 and 23.66 µM acetate, respectively (Figure 4C). *OR51E2* siRNA knockdown in the MZ-CRC-1 cell line reduced C^11^-acetate uptake (18.08% ± 4.2%) significantly (*p* < 0.05) when compared to the levels in the parental cell. Similarly, there was a significant reduction in C^11^-acetate uptake in *Olfr78*-KO mice compared to that in WT mice (51.79%, *p* = 0.026, Figure 4D).

Finally, we confirmed C^11^-acetate accumulation in a 31-year-old male patient with MTC who underwent C^11^-acetate PET/CT examinations to evaluate suspected metastasis (serum calcitonin: 320 pg/mL, serum CEA: 27 ng/mL). C^11^-acetate PET/CT revealed increased C^11^-acetate levels in the primary lesion and mediastinal lymph node metastasis (Appendix A).

## 4. Discussion

Herein, we demonstrate that acetate-OR51E2 interaction induces calcitonin secretion through the cAMP pathway in C-cells of the thyroid gland and is a potential MTC bio-marker. We investigated whether OMP is expressed in C-cells of human patients. We found that calcitonin-secreting cells, such as C-cells and MTC cells, express OR51E2. This is consistent with previous reports that ORs play functional roles in non-olfactory tissues [4,5,12,17,18]. The role of calcitonin, in addition to its functions in bone metabolism, has not been clearly defined. We have not focused on the clinical significance of the difference between OR51E2 and calcitonin excretion but instead on its potential application in the identification of C-cells within MTC. The biomarker might address the current need to locate MTC recurrence in patients with elevated calcitonin levels [19].

Extranasal odorant receptors, which belong to the family of G-protein-coupled receptors (GPCRs), have been discovered in various tissues and organs outside of the olfactory system, suggesting their diverse functions [3,7,9,10,11,20,21]. These odorant receptors go beyond their role in odor perception and play important roles in different physiological processes. In the gastrointestinal tract, they are involved in detecting nutrients and regulating digestive functions such as nutrient absorption and hormone secretion. In the respiratory system, odorant receptors contribute to the recognition of airborne chemicals and pollutants, playing a role in respiratory defense mechanisms and airway inflammation. In the male and female reproductive systems, odorant receptors have been identified, indicating potential involvement in reproductive processes and fertility. Within the immune system, odorant receptors participate in recognizing and responding to microbial pathogens, influencing immune cell activation and inflammatory responses. Additionally, odorant receptors in the heart and blood vessels may contribute to the regulation of cardiovascular functions, including blood pressure and cardiac contractility. In the urinary system, odorant receptors are implicated in detecting chemical components in urine, potentially influencing renal function and fluid balance. Odorant receptors have also been found in adipose tissue, suggesting possible roles in energy metabolism and adipocyte function. Moreover, extranasal odorant receptors are being investigated for their potential involvement in cancer development and progression, as they have been found in various cancer cells and may contribute to tumor growth and metastasis. Considering the diverse expression of odorant receptors in all of these organs, we surmise that our results may be helpful in finding the function and expression of these odorant receptors in various pathological conditions.

We found that approximately 20 µM acetate displaces half of the bound C^11^-acetate from the OR51E2 receptor in vitro. Considering the specific activity of C^11^-acetate, only a small amount of C^11^-acetate is required for receptor imaging using PET, which suggests that C^11^-acetate PET/CT could be applied in clinical settings in the location of calcitonin-secreting cells, which express OR51E2. Overall, the results suggested that the acetate-OR51E2 interaction increased calcitonin secretion through the cAMP pathway in C-cells of the thyroid gland. One caveat is that acetate may not be specific in its activity to only target OR51E2. Previous reports have suggested that GPCR41/43 is another major acetate target receptor involved in adipogenesis and insulin secretion [13]. Collectively, the data suggest that acetate-Olfr78 induces calcitonin secretion in C-cells of the thyroid gland in animals.

Although we did not directly evaluate whether the observed C^11^-acetate signal was due to GPCR41/43 binding, acetate response diminished following OR51E2 knockdown, suggesting that the acetate response occurs mainly through OR51E2 in C-cells. Due to the extremely low prevalence of MTC and infrequent synthesis of C^11^-acetate, further human studies were difficult. Because of the interchangeability of PET radioisotopes, acetate may be a suitable carrier for alpha or beta-emitting radioisotopes, which could facilitate the identification of MTC remnants following surgery or treatment of MTC and distant metastasis in patients who do not respond to conventional therapy.

In conclusion, we demonstrated that acetate is a ligand for OR51E2 expressed in MTC and could be capitalized on in diagnostic and therapeutic methods. We recommend the application of C^11^-acetate PET/CT targeting OR51E2 as a novel biomarker for identifying recurrent sites in MTC diagnosis for patients who have undergone thyroidectomy.

## Figures and Tables

**Figure 1 biomedicines-11-01688-f001:**
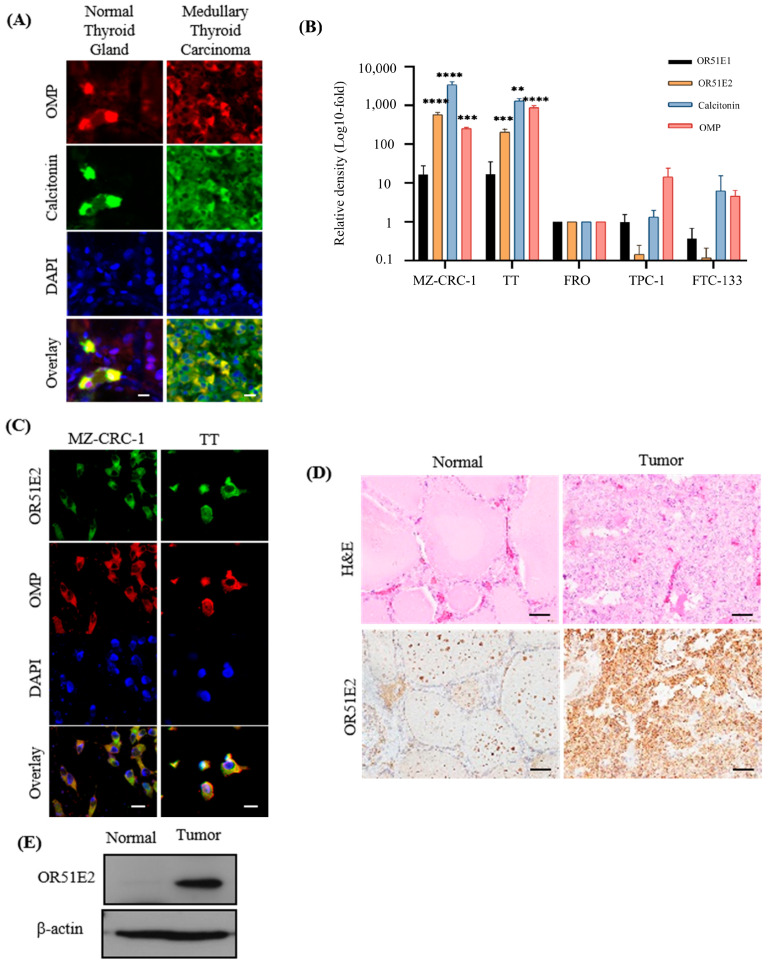
C-cell-specific OR51E2 expression increases in medullary thyroid cancer. (**A**) Immunofluorescence staining of calcitonin (green) and olfactory marker protein (OMP, red) in normal human thyroid and medullary thyroid cancer tissue (scale bar: 10 µm). (**B**) mRNA expression levels of OR51E2, OMP, calcitonin, and OR51E1 were measured in medullary and anaplastic/papillary/follicular cancer cell lines. *GAPDH* was used to normalize expression, and relative expression was confirmed using the 2^−ΔΔCt^ method based on FRO values. Data are presented as relative density (fold) compared to FRO cell lines. One-way ANOVA, *n* ≥ 3, (** *p* < 0.01; and *** *p* < 0.005; **** *p* < 0.001). (**C**) Immunofluorescent staining demonstrated co-localization of OR51E2 (green) and OMP (red) in the MZ-CRC-1 and TT (medullary thyroid cancer [MTC] cell lines) (scale bar: 10 µm). (**D**,**E**) Immunohistochemical staining (**D**) and Western blotting analysis (**E**) revealed that OR51E2 was expressed in normal thyroid and MTC tissue in a 31-year-old patient with MTC (scale bar: 10 μm).

**Figure 2 biomedicines-11-01688-f002:**
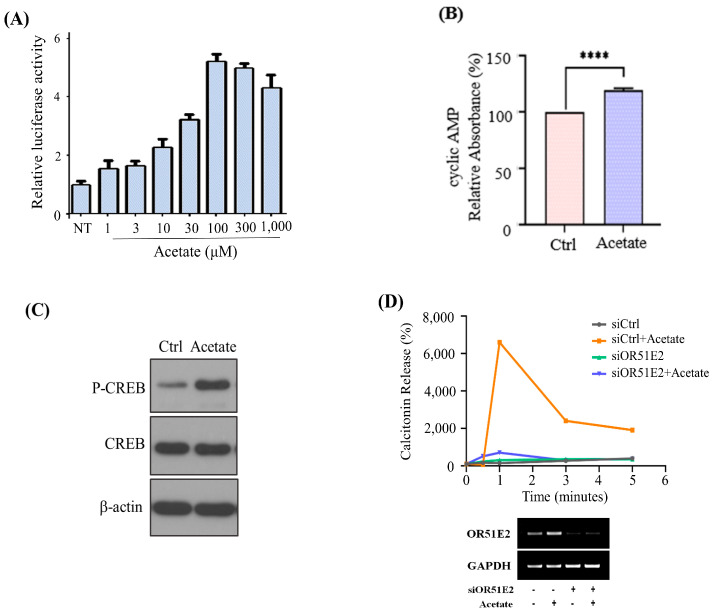
Acetate-stimulated OR51E2 increases calcitonin secretion through the cAMP pathway. (**A**) Relative luciferase activities of OR51E2 vs. concentrations of the indicated acetate in MZ-CRC-1 cells. Data are expressed as mean ± standard error of mean (SEM) (*n* = 5) at each concentration. (**B**–**D**) Acetate-OR51E2 induced calcitonin secretion through the cAMP pathway. (**B**) cAMP levels were measured in MZ-CRC-1 cells 1 h after 250 µM acetate treatment. All data are expressed as mean ± SEM. ****, *p* < 0.0001 by one-way ANOVA with Student’s *t*-test. (**C**) Western blot with anti-phospho-CREB antibody showing the phosphorylation of CREB in the MZ-CRC-1 cells 15 min after acetate treatment. β-actin was used as the loading control. (**D**) OR51E2 siRNA or non-targeting siRNA (siCtrl) was transfected into MZ-CRC-1 cells for 48 h. Medium was collected in a time-dependent manner (0, 1, 2, 3, 4, and 5 min) after 250 µM acetate treatment, and the calcitonin levels were measured using an ELISA kit. RT-PCR analysis of OR51E2 mRNA expression in MZ-CRC-1 cells transfected with siCtrl or siOR51E2 (bottom).

**Figure 3 biomedicines-11-01688-f003:**
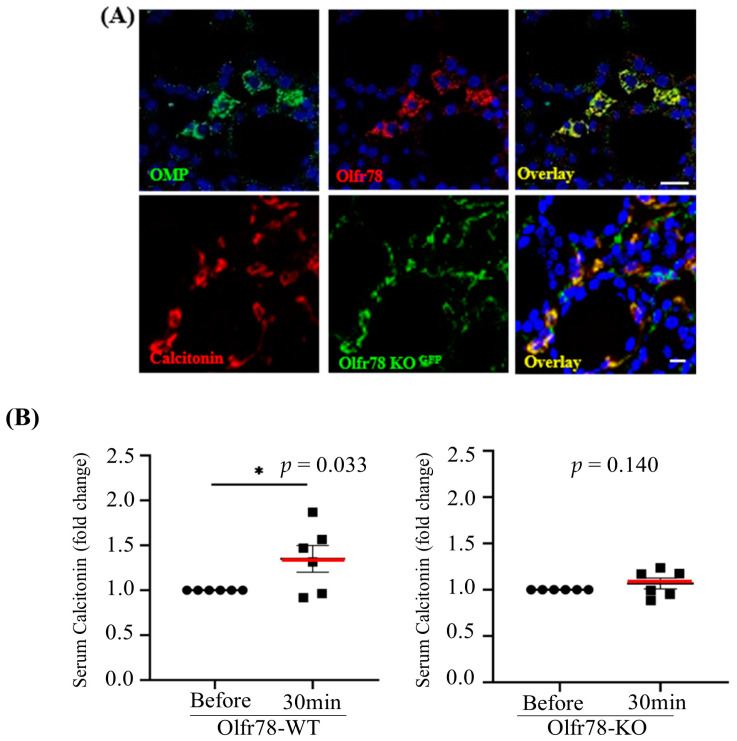
Acetate-Olfr78 binding increases serum calcitonin secretion in the animal model. (**A**) Immunofluorescence showing co-localization of the odorant receptor Olfr78 and calcitonin with OMP in C57/BL6 mouse thyroid gland. The upper panels show the double labeling of OMP and Olfr78 using anti-OR51E2 (Olfr78 human ortholog: red) and anti-OMP (green). GFP-replaced *Olfr78* knockout (KO) mice revealed co-localization of calcitonin and GFP signal in the normal thyroid gland (bottom panels) (scale bar: 10 μm). (**B**) Increased calcitonin secretion from acetate-injected wild-type (WT) mice decreased in *Olfr78*-KO mice—intraperitoneal injection of acetate (500 mg/kg) was performed in six-week-old WT or *Olfr78*-KO mice. After 30 min, serum was isolated from the blood obtained via retro-orbital blood collection, and calcitonin levels were measured. All data are expressed as mean ± SEM (*n* = 6). * *p* < 0.05 using one-way ANOVA and a paired *t*-test.

**Figure 4 biomedicines-11-01688-f004:**
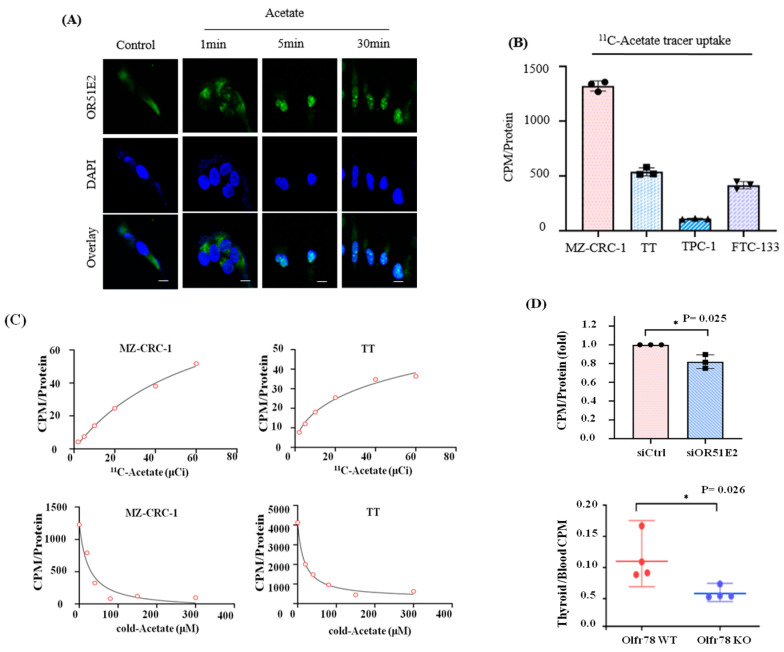
Acetate is internalized with OR51E2 in the MTC. (**A**) Nuclear translocation of OR51E2 after acetate treatment in MZ-CRC-1 cells—serum-starved MZ-CRC-1 cells were incubated with acetate at indicated times, and cells were immunostained with anti-OR51E2 antibody (scale bar: 10 μm). (**B**) C^11^-acetate uptake assay in thyroid cancer cell lines revealed the highest uptake in MZ-CRC-1 cells. The cells were treated with C^11^-acetate (at 5 µCi) for 10 min; they were collected and counted using a gamma counter. The results were normalized with protein concentration. Mann–Whitney test, *p* < 0.001 for MZ-CRC-1 vs. all the other cell lines, *p* < 0.001 for TT vs. TPC-1, and *p* = 0.038 for TT vs. FTC-133 (*n* = 3). (**C**) Dose-dependent C^11^-acetate uptake in the medullary thyroid cancer cell lines (top panels). The cells were treated with C^11^-acetate (2, 5, 10, 20, 40, and 60 µCi) for 10 min. For the C^11^-acetate competitive binding assay (bottom panels), non-radioactive acetate treatment was performed for 3 min in a dose-dependent manner. After PBS washing, 10 µCi of C^11^-acetate treatment for 10 min was performed. Cells were collected, counted using a gamma counter, and normalized using the protein concentration. (**D**) In vitro assay (up) C^11^-acetate uptake in OR51E2 siRNA knockdown MZ-CRC-1 cells decreases significantly. After transfection of MZ-CRC-1 cells with non-targeting siRNA (siCtrl) or siOR51E2 for 72 h, cells were incubated with 10 µCi C^11^-acetate for 10 min, and the C^11^-acetate uptake was measured. Cells were harvested, counted using a gamma counter, and counts in the samples were normalized with intracellular protein concentration. All data are means ± SEM (*n* = 3). * *p* < 0.05 using one-way ANOVA with a paired *t*-test. In vivo radiotracer assay (bottom) C^11^-acetate uptake decreased in *Olfr78*-KO mice compared to WT-tumor-bearing mice (*n* = 4). C^11^-acetate uptake was confirmed in eight-week-old *Olfr78*-KO and WT mice. C^11^-acetate was injected into the tail vein, and the mice were sacrificed 10 min later. The C^11^-acetate amounts in thyroid gland and orbital blood were counted using a gamma counter and normalized using blood values. All the data are means ± SEM (*n* = 3). * *p* < 0.05 using one-way ANOVA and the Mann–Whitney test.

## Data Availability

The datasets are available upon reasonable request to the corresponding author.

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
