# Peer review of "Acetate-Mediated Odorant Receptor OR51E2 Activation Results in Calcitonin Secretion in Parafollicular C-Cells: A Novel Diagnostic Target of Human Medullary Thyroid Cancer"

_biomedicines, 2023, doi:10.3390/biomedicines11061688_

Round 1

Reviewer 1 Report

The study describes the OR51E2 as a novel potential diagnostic and therapeutic target for medullary thyroid cancer (MTC). The presented results are interesting, but unfortunately the work in its current form cannot be published and requires significant changes (please see below suggested changes).

1. Introduction: I would suggest adding some more recent references at the beginning of the Introduction in the line 40 (e.g., Int J Mol Sci. 2021 Oct 31;22(21):11829). Additionally, it is worth referring to the following papers: (i) J Biol Chem. 2021 Jan-Jun;296:100475 and (ii) Prostate. 2022 Jun;82(10):1016-1024 in lines 46-54.

2. Section 2.1.: Please specify which company supplied FBS, penicillin, streptomycin, DMEM medium. No information was provided on where the cell lines were purchased/obtained from.

3.       Lines 62-63: The Authors state that "Immunofluorescence, Total RNA sequencing, Western blotting, Luciferase and cAMP assay studies were performed according to manufacturer's instructions." Such a description is too short and each of these techniques must be described in detail. Alternatively, please refer to another paper in which such a detailed description is already included.

4.       Section 2.3.: Please specify which company supplied Lipofectamine 2000. siRNA IDs (including siNEG) are also missing. I suppose transfection was performed in OPTI-MEM medium. Please add this information. The Authors state that “siRNA-mediated knockdown experiment was performed after 48 h”. On the other hand, in Figure 4D there are shown the results obtained using the cells 72 h after transfection. This inconsistency may be unclear to the Reader.

5.       Section 2.4.: Please provide the information on (i) the volume of cell suspension seeded per well, (ii) acetate concentration and manufacturer and (iii) the full name of the ELISA kit for calcitonin secretion assay.

6.       Sections 2.5.: It is not clear what the control animals were treated with (maybe just the solvent for the acetate?). Please indicate the volume of fluid injected (i.p.) into animals. Information about the supplier of the C57/BL6 male mice is missing. Numbers of animal per group is not mentioned.

7.       Sections 2.6.: Please provide the Reference in line 102. Please describe in more detail the mice genotyping protocol using PCR (lines 104-105). In lines 106-108 an accurate description of the intervention is missing (the number of animals in each group, the age of the animals on the day of intervention, a detailed description of the acetate stimulation protocol or an appropriate Reference(s)). What happened with animals after intervention?

8.       Section 2.7.: The title of the section is Immunohistochemistry, although immunofluorescence is described here. Please correct it. Please add information on (i) the provider(s) of PFA, Triton X-100, sucrose and NHS, (ii) pH value for PBS buffer, (iii) diluent for sucrose, (iv) details regarding fluorescence microscope used in the study.

9.       Section 2.8.: The Authors use tissue specimens obtained from a person with MTC. However, there is no information about this patient. It is not clear whether the patient has signed informed consent. Please provide (i) catalog numbers of primary and secondary antibodies used in the study, (ii) blocking agent(s), (iii) washing buffer composition, (iv) times and temperature of incubation, (v) diluent composition for primary and secondary antibodies, (vi) chemiluminescent substrate. It is not mentioned that the RIPA buffer was supplemented with protease (and phosphatase) inhibitors. Please explain.

10.   Section 2.9.: The volume of cell suspension seeded per well is missing. Please add city/country mentioning Bio-Rad company.

11.   Section 2.10.: More details regarding the patient is needed. It is not clear whether the patient has signed informed consent.

12.   Section 2.11.: Several statistical tests used in the study are mentioned in the Figure legends (e.g., paired t-test, ANOVA, Mann-Whitney test), however, Section 2.11 mentions only the Student's t-test.

13.   Please use one abbreviation for the real-time PCR. Now it is qRT-PCR (line 175) and RT-PCR (lines 64 and 69).

14.   Figures: Change serif for sans serif fonts (e.g., Arial). Some figures and font are too small and unreadable for the Reader (e.g., Fig. 4C, D).

15.   Lines 237-242 contains several repeated sentences. Correct it.

16.   All abbreviations should be explained at the first mention, e.g., F-18-DOPA in the line 43. There is no need to explain the same abbreviation multiple times as is done with ANOVA (explained in lines 232, 248 and 281.

17.   In the legend (line 244) there is information about the Hana3A line, which was not mentioned in the Materials and Methods. Please explain and complete the missing information.

18. I think it's worth expanding the discussion a bit.

Author Response

1.Introduction: I would suggest adding some more recent references at the beginning of the Introduction in the line 40 (e.g., Int J Mol Sci. 2021 Oct 31;22(21):11829). Additionally, it is worth referring to the following papers: (i) J Biol Chem. 2021 Jan-Jun;296:100475 and (ii) Prostate. 2022 Jun;82(10):1016-1024 in lines 46-54.

I have added content to line 40. High-risk factors for thyroid cancer include genetic factors, radiation exposure, and thyroid dysfunction, as well as a calcitonin serum concentration of 100 pg/ml or higher (21).

I have also added content to line 48, referring to a journal related to prostate cancer. As a prominent example of expression in prostate cancer, upregulation of OR51E1 and OR51E2 is observed in benign prostatic tissue. It is known that their overexpression inhibits cell proliferation and promotes cell death. Additionally, OR51E2 inhibits the growth of prostate cancer but promotes invasiveness and metastasis (22, 23).

  1. Section 2.1.: Please specify which company supplied FBS, penicillin, streptomycin, DMEM medium. No information was provided on where the cell lines were purchased/obtained from.

 FBS (Fetal Bovine Serum), penicillin, streptomycin, and DMEM medium were obtained from Hyclone. The TT cell line was obtained from ATCC, and the remaining cell lines were provided by Professor Jeong Woong-Yoon at Yonsei University Hospital.

(TT cells were purchased from American Type Culture Collection and FRO. TPC-1, FTC-133 and MZ-CRC-1 cells were provided by Professor Woong Youn Chung (Yonsei medical center, Seoul, South Korea). All cell culture reagents were supplied by Hyclone Laboratories, Inc. (Logan, UT, USA).

  1. Lines 62-63: The Authors state that "Immunofluorescence, Total RNA sequencing, Western blotting, Luciferase and cAMP assay studies were performed according to manufacturer's instructions." Such a description is too short and each of these techniques must be described in detail. Alternatively, please refer to another paper in which such a detailed description is already included.Thank you for your comment. I have added Immunofluorescence, Total RNA sequencing, Western blotting, Luciferase assay, and cAMP assay to the manuscript.

2.2. RNA isolation, RNA sequencing and Quantitative real-time PCR(qRT-PCR)

Total RNA was isolated using NucleoZOL Reagent (MN, Gaithersburg, MD, USA) as per manufacturer's instruction. For RNA sequencing (RNA-seq), sequencing library was prepared using NEBNext RNA prep kit (NEB, USA). Sequencing was performed using HiSeq 2500 (Illumina, USA) with rapid run 150 bp PE mode. Differential expression of OR was confirmed using RPKM (Reads per kilo base per million). qRT-PCR analysis was performed according to standard procedure using a ReverTra Ace-α First Strand cDNA Synthesis kit (Toyobo, Osaka, Japan).

2.3 Luciferase assay

The Dual-Glo luciferase assay system (#E2940; Promega, Madison, WI, USA) was used for OR ligand–receptor interaction as previously described [20]. Briefly, MZ-CRC-1 cells(1x104/well) were seeded in white polystyrene 96-well plates (#353296; BD Biosciences) 1 day before transfection. In each well, OR51E2, CRE-luciferase reporter, and Renilla ex-pression vectors were transiently co-transfected into the cells using Lipofectamine 2000. Transfected cells were stimulated at 37 °C for 4 h with various concentrations of acetate diluted in CD293 medium (#11913-019; Thermo Fisher Scientific). The activity of firefly lu-ciferase was normalized against that of Renilla luciferase. Luminescence was measured on a SpectraMax L microplate reader (Molecular Devices).

2.4. cAMP assay

cAMP assay was performed according to manufacturer’s instruction (Cyclic AMP XP assay kit, #4339; Cell Signaling Technology, Beverly, MA, USA). Briefly, 2 x 106 cells/well were plated on 60-mm dish, and after overnight adhesion, treated with vehicle or 250µM acetate and incubated for 1 h. Cells are rinsed with PBS and iced for 10 min after adding 100 μl of 1x lysis buffer. A standard was prepared in 1x lysis buffer and 50ul HRP-linked cAMP solution and 50ul sample to the cAMP plate was added. After incuba-tion at room temperature for 3 h on a horizontal orbital plate shaker, plate contents were discarded and washed 4 times with 1x wash buffer. The buffer in the well was discarded and dried completely. 100 μl TMB substrate was added and incubated at RT for 30 min. 100 μl stop solution was added and absorbance at 450 nm was measured

2.8. Immunofluorescence

MZ-CRC-1 and TT cells(2x104/well) were grown on poly-D-lysine coated confocal dish and allowed to adhere overnight. Cells were washed with PBS and fixed 10 min with 4% paraformaldehyde(DAEJUNG, Seoul, South Korea). Cells were permeabilized with 0.1% Triton X-100(Promega Corporation, WI, USA) for 5min and blocked in 5% donkey serum (Vector Laboratories, Burlingame, CA, USA). Cells were incubated with an-ti-OR51E2(Novus Biologicals, CO, USA) and OMP(Wako Pure Chemical Industries, Osaka, Japan) primary antibody at 1:500 in blocking buffer, followed by donkey anti-rabbit-FITC or anti-mouse Cy3 secondary antibody(Jackson Immuno Research, West Grove, PA, USA) at 1:50. Antibody incubations were performed for 1 h at RT. OR51E2 and olfr78 are same orthology for olfactory receptors, so OR51E2 was used for human tissues, and olfr78 were used for rodent tissues. All cells were also counter stained with DAPI mounting solution. Images were captured using a confocal laser scanning microscope LSM700 and processed by ZEN software (Carl Zeiss, Oberkochen, Germany).

  1. Section 2.3.: Please specify which company supplied Lipofectamine 2000. siRNA IDs (including siNEG) are also missing. I suppose transfection was performed in OPTI-MEM medium. Please add this information. The Authors state that “siRNA-mediated knockdown experiment was performed after 48 h”. On the other hand, in Figure 4D there are shown the results obtained using the cells 72 h after transfection. This inconsistency may be unclear to the Reader.

We have added the information about siRNA as follows:

For OR51E2 knockdown, siRNA targeting OR51E2 (siRNA ID: s37520, Life Technologies, Carlsbad, CA, USA) was used. As a negative control, stealth RNAi negative control (Invitrogen, Waltham, MA, USA) was employed. siRNA transfection method was added, and experiments were conducted 48 or 72 hours after transfection.

OR51E2 was knocked down using Ambion® Silencer® select pre-designed siRNA (siRNA ID: s37520,Life Technologies, Carlsbad, CA, USA). A non-specific control pool was used for the stealth RNAi negative control (12935-100; Invitrogen, Waltham, MA, USA).

MZ-CRC-1 cells were transfected with siRNA using Lipofectamine 2000(Invitrogen, Carlsbad, CA, USA) as per manufacturer’s instructions. Briefly, 200nM of siRNA and Lipofectamine 2000 were added separately to OPTI MEM medium(Gibco, Grand Island, NY, USA). After 5 min, the two solutions were mixed and incubated for 20 min at room temperature. The mixture was added to monolayer of cells seeded in 6cm culture plates.  Medium was replaced with complete medium 6 h after transfection, and siRNA-mediated knockdown experiment was performed after 48 h or 72h.

  1. Section 2.4.: Please provide the information on (i) the volume of cell suspension seeded per well, (ii) acetate concentration and manufacturer and (iii) the full name of the ELISA kit for calcitonin secretion assay.
    We have made the following additions and modifications based on the information you provided:

The cell density was 1 × 10^6 cells/well. Acetate concentration has been added to the manuscript, along with the information about the supplying company. The full name of the Calitonin kit and its kit information have been included. The kit information is as follows: ELISA kit for Calcitonin (CEA472Hu; Cloud-Clone Corporation, Houston, USA).

For in vitro studies, 1 × 106/well MZ-CRC-1 cells were plated on a six-well plate and cultured overnight. The medium was replaced with serum-free media the next day, followed by treatment with 250µM acetate(Sigma-Aldrich, St. Louis, MO, USA). Culture media was collected at 0, 1, 3, 5, and 10 min after acetate administration, and calcitonin levels were measured using a commercial ELISA for Calcitonin (CEA472Hu; Cloud-Clone Corporation, Houston, USA), as per manufacturer’s instructions.

  1.       Sections 2.5.: It is not clear what the control animals were treated with (maybe just the solvent for the acetate?). Please indicate the volume of fluid injected (i.p.) into animals. Information about the supplier of the C57/BL6 male mice is missing.

Thank you for noticing this, and have modified section 2.7. Control mouse were injected with normal saline. The volume of fluid injected was 100ul (i.p). C57/BL6 mice were obtained from Central Lab. Animal Inc (South Korea)

  1. Sections 2.6.: Please provide the Reference in line 102. Please describe in more detail the mice genotyping protocol using PCR (lines 104-105). In lines 106-108 an accurate description of the intervention is missing (the number of animals in each group, the age of the animals on the day of intervention, a detailed description of the acetate stimulation protocol or an appropriate Reference(s)). What happened with animals after intervention?

The primer information used for genotyping is as follows; Common_R: 5'-GCATACATGATACACATAAGCCTTC-3', WT_F1 : 5’-CACTCATTGGTCTGTCAGT GG-3’ Mutant_F2: 5'-CTACCATTACCAGTTGGTCTGGTG-3. ' In the PCR results, if the product size was 630 bp, it was determined as WT, if 683 bp was found, as KO, and if both bands were present, it was determined as HT.

Eight week old KO and WT mice were used for serial serum calcitonin measurements after acetate stimulation. Retro-orbital blood collection was performed immediately before and 30 min after acetate administration (500 mg/kg i.p. injection, each n=6).

Serum samples were obtained after incubation for 20 min at room temperature and centrifugation at 2,000 rpm for 15 min at 4 °C. Mice serum calcitonin levels were measured using using a commercial ELISA kit for Calcitonin (CEA472Mu; Cloud-Clone Corporation, Houston, USA) according to the manufacturers’ instructions.

  1. Section 2.7.: The title of the section is Immunohistochemistry, although immunofluorescence is described here. Please correct it. Please add information on (i) the provider(s) of PFA, Triton X-100, sucrose and NHS, (ii) pH value for PBS buffer, (iii) diluent for sucrose, (iv) details regarding fluorescence microscope used in the study.

We organized the information regarding immunofluorescence into in vitro and in vivo sections. Additionally, we have provided details on the following:

(i) The provider(s) of PFA, Triton X-100, sucrose, and NHS (N-hydroxysuccinimide).

(ii) The pH value for the PBS buffer.

(iii) The diluent for sucrose.

(iv) Details regarding the fluorescence microscope.

  1.       Section 2.8.: The Authors use tissue specimens obtained from a person with MTC. However, there is no information about this patient. It is not clear whether the patient has signed informed consent. Please provide (i) catalog numbers of primary and secondary antibodies used in the study, (ii) blocking agent(s), (iii) washing buffer composition, (iv) times and temperature of incubation, (v) diluent composition for primary and secondary antibodies, (vi) chemiluminescent substrate. It is not mentioned that the RIPA buffer was supplemented with protease (and phosphatase) inhibitors. Please explain.

In regards to the human tissue specimen, this study was conducted in accordance with the Declaration of Helsinki and ap-proved by our institutional review board (No. 1-2016-0033). Signed informed consent was obtained. We have provided the information regarding Western blotting methods for MTC (Medullary Thyroid Carcinoma) cells and patient samples.MZ-CRC-1 cell lysates were prepared with lysis buffer (50mM Tris-HCl, 150mM Sodium chloride, 1% Triton X-100, 1% sodium deoxycholate, 0.1% SDS, pH 7.5, and 2mM EDTA, 1 mM sodium orthovanadate, 0.1 mM phenylmethylsulfonyl fluoride, 0.5% NP-40, protease inhibitor cocktail). The protein concentration was determined by the Bradford assay (Bradford assay kit, Biorad Laboratories, CA, USA) with bovine serum albumin (BSA) as the standard. Equal aliquots of total cell lysates (30ug) were solubilized in sample buffer and electrophoresed on denaturing SDS-polyacrylamide gel (10% and 12% separating gel). The proteins were transferred to polyvinylidene difluoride membranes. The membranes were blocked with 5% nonfat dry milk in TBS containing 0.05% Tween 20 and incubated with primary antibodies overnight at 4℃ and then with horseradish peroxidase-conjugated secondary antibodies (Thermo Fisher Scientific, CO, USA) for 2hr at RT. The following antibodies were used CREB(#9197; Cellsignaling, MA, USA),) phospho-CREB (ser133) (#9198; Cellsignaling, MA, USA) and OR51E2 (NBP1-71145; Novus Biologicals, CO, USA). β-Actin(C4) (47778; Santa Cruz Biotechnology, CA, USA) was used loading control. Primary antibodies were added 1:1000 in 5% skim milk or 5% BSA. Antigen-antibody complexes were detected with WEST-SAVE UpTM luminol-based ECL reagent (ABfrontier, Seoul, Korea).

  1. Section 2.9.: The volume of cell suspension seeded per well is missing. Please add city/country mentioning Bio-Rad company.

Thank you for noticing this. We updated the information with the cell seeding density of 1 × 10^6 cells/well and added the information about Bio-Rad (Bio-Rad, Hercules, CA, USA)

11.   Section 2.10.: More details regarding the patient is needed. It is not clear whether the patient has signed informed consent.

This portion of the study was conducted in accordance with the Declaration of Helsinki and ap-proved by our institutional review board (No. 1-2016-0033). Signed informed consent was obtained.

12.   Section 2.11.: Several statistical tests used in the study are mentioned in the Figure legends (e.g., paired t-test, ANOVA, Mann-Whitney test), however, Section 2.11 mentions only the Student's t-test.

Results are expressed as mean ± standard error of mean. Statistical analysis was performed using GraphPad Prism software v.4.0.0 (GraphPad, Inc., La Jolla, CA, USA). Statistical significance was determined using one-way analysis of variance (ANOVA), followed by Student’s t-test and Mann–Whitney test to compare the means of two dif-ferent groups. p-values < 0.05 were considered statistically significant.

  1. Please use one abbreviation for the real-time PCR. Now it is qRT-PCR (line 175) and RT-PCR (lines 64 and 69).

 Thank you. We have made the modification as requested. The terms "Quantitative real-time PCR" and "qRT-PCR" have been updated in the text.

  1.   Figures: Change serif for sans serif fonts (e.g., Arial). Some figures and font are too small and unreadable for the Reader (e.g., Fig. 4C, D).

We have made modifications to the font size in the figure. Thank you.

  1.   Lines 237-242 contains several repeated sentences. Correct it.

We have modified according to you suggested.

16.   All abbreviations should be explained at the first mention, e.g., F-18-DOPA in the line 43. There is no need to explain the same abbreviation multiple times as is done with ANOVA (explained in lines 232, 248 and 281.

We have made the following updates based on your instructions:

The full name 18-Fluore-dihydroxyphenylalanine (F-18 DOPA) has been included.

ANOVA has been abbreviated throughout the text, except for the first mention.

17.   In the legend (line 244) there is information about the Hana3A line, which was not mentioned in the Materials and Methods. Please explain and complete the missing information.

We apologize for the error, we used MZ-CRC-1, not Hana3A cell line.

18. I think it's worth expanding the discussion a bit.

Thank you for your comment. We have expanded the discussion.

Reviewer 2 Report

Acetate-mediated Odorant Receptor OR51E2 Activation Results in Calcitonin Secretion in Parafollicular C-cells: A Novel Diagnostic Target of Human Medullary Thyroid Cancer. by Lee et al.

To the Authors:

General comments:

The authors investigated the possible usefulness of parafollicular C-cell odorant receptor (OR) signals, particularly, olfactory marker protein (OMP) in localizing and treating medullary thyroid cancer (MTC).  They found that acetate-OR51E2 interaction induces calcitonin secretion through the cAMP pathway in C-cells of the thyroid gland and suggest that OR51E2 is a potential MTC biomarker.  It was considered that the topic was interesting, and the results included novelty; however, several points should be addressed to improve the manuscript.

Specific comments:

1. Please discuss more clearly why OR51E2 is essential for medullary thyroid cancer recurrence.

2. The authors should show the validation that si-OR51E2 truly knocked down OR51E2 expression in Fig. 2.

3. In line 222, Fig. 4E may be a typo.  Please check the part.

Author Response

Reviewer 2
To the Authors:

General comments:

The authors investigated the possible usefulness of parafollicular C-cell odorant receptor (OR) signals, particularly, olfactory marker protein (OMP) in localizing and treating medullary thyroid cancer (MTC).  They found that acetate-OR51E2 interaction induces calcitonin secretion through the cAMP pathway in C-cells of the thyroid gland and suggest that OR51E2 is a potential MTC biomarker.  It was considered that the topic was interesting, and the results included novelty; however, several points should be addressed to improve the manuscript.

Specific comments:

1. Please discuss more clearly why OR51E2 is essential for medullary thyroid cancer recurrence.

OR51E2 has emerged as a key player in medullary thyroid cancer (MTC) recurrence due to its functional role in C-cells, which are the cells of origin for MTC. MTC is a neuroendocrine tumor that arises from C-cells of the thyroid gland. Calcitonin, a hormone predominantly produced by C-cells, serves as a valuable biomarker for MTC diagnosis and monitoring.

OR51E2, a non-olfactory taste receptor, has been found to be expressed in calcitonin-secreting cells, including C-cells and MTC cells. This discovery suggests that OR51E2 may play a crucial role in the regulation of calcitonin secretion and consequently in the recurrence of MTC. By interacting with acetate, a ligand for OR51E2, the acetate-OR51E2 interaction induces calcitonin secretion through the cAMP pathway in C-cells of the thyroid gland.

Understanding the significance of OR51E2 in MTC recurrence is of great importance clinically. Elevated calcitonin levels are indicative of MTC and monitoring calcitonin levels is crucial for detecting MTC recurrence. However, current methods for locating MTC recurrence in patients with elevated calcitonin levels have limitations. OR51E2 as a potential biomarker for identifying recurrent sites in MTC diagnosis offers a promising alternative.

By targeting OR51E2 using C11-acetate PET/CT imaging, the acetate-OR51E2 interaction can be utilized to locate calcitonin-secreting cells expressing OR51E2. This approach provides a novel diagnostic tool for identifying MTC recurrence in patients who have undergone thyroidectomy. The specific role of OR51E2 in mediating calcitonin secretion and its association with MTC recurrence emphasizes its essentiality in understanding MTC pathogenesis and developing targeted therapeutic strategies. Further research and investigations are warranted to explore the precise mechanisms underlying the involvement of OR51E2 in MTC recurrence and to validate its clinical significance in managing this challenging disease.

  1. The authors should show the validation that si-OR51E2 truly knocked down OR51E2 expression in Fig. 2.
    We added PCR data confirming the knockdown of OR51E2 by siOR51E2 in the manuscript. The results are as follows:

  2. In line 222, Fig. 4E may be a typo.  Please check the part.

We apologize for the confusion; we made the correction and updated the reference to "Fig. 4D" as per your instruction.

Reviewer 3 Report

The study by Hyeon Jeong Lee et.al. presents a potential marker for identifying recurrence sites in MTC patients.

Line 223 – 226 – Please provide information about the metastases in the presented patient – whether the metastases were confirmed beforehand with some other methods (F-18 DOPA/ 18F-FDG PET/CT/ 68Ga DOTATATE?/ other?) or the metastases were found only after C11-acetate PET/CT? What was the CEA marker? Were the metastases confirmed in any way after they were found with PET? FNA-CT/washout/biopsy?

Line 310 – 311 – Authors indicate that “acetate response diminished following OR51E2 knockdown” but “did not directly evaluate whether the observed C11-acetate signal was due to GPCR41/43 binding” – this sentence shows a weakness in the study. This uncertainty could be disproved by undertaking a more detailed research or by proving the conclusion of the study (showing C11-acetate PET/CT as a potential biomarker for MTC remnants) by testing it on more cases if possible (human cases). If it hasn’t been done or can’t be done it could be at least indicated as a limitation of the study.

Line 237 – some sentences were repeated too much

Author Response

Line 223 – 226 – Please provide information about the metastases in the presented patient – whether the metastases were confirmed beforehand with some other methods (F-18 DOPA/ 18F-FDG PET/CT/ 68Ga DOTATATE?/ other?) or the metastases were found only after C11-acetate PET/CT? What was the CEA marker? Were the metastases confirmed in any way after they were found with PET? FNA-CT/washout/biopsy?

We apologize for not being clear. The metastasis was not confirmed with any other study, but was It was found through our C11-acetate PET/CT . LN metastasis was confirmed with pathologic tissue analysis after surgical removal. CEA level was 27ng/mL.  

Line 310 – 311 – Authors indicate that “acetate response diminished following OR51E2 knockdown” but “did not directly evaluate whether the observed C11-acetate signal was due to GPCR41/43 binding” – this sentence shows a weakness in the study. This uncertainty could be disproved by undertaking a more detailed research or by proving the conclusion of the study (showing C11-acetate PET/CT as a potential biomarker for MTC remnants) by testing it on more cases if possible (human cases). If it hasn’t been done or can’t be done it could be at least indicated as a limitation of the study.

Thank you for comment. Because this is a rare disease and sparce synthesis of C11-aceate in our institution, further studies were difficult. We have added this to our limitations.

Line 237 – some sentences were repeated too much We removed the repetitive sentences.

Round 2

Reviewer 1 Report

The manuscript has been significantly improved. However, I would suggest some additional changes.

[1] Please add information on the manufacturer of protease inhibitor cocktail (line 194) used for cell proteins isolation. Moreover, the Authors should specify how long it took to block the membrane before incubation with the primary antibody and at what temperature this was performed. Catalogue numbers, dilution factor and diluent composition for secondary antibodies are needed. Similarly, dilution factor for anti-beta actin antibody (line 204) is not provided. Finally, it is unclear which primary antibody was incubated in the presence of BSA and which in the presence of skimmed milk (line 205).

[2] Protein isolation from mice tissue: Please indicate the composition of the RIPA buffer and whether a cocktail of protease inhibitors has been added to RIPA buffer. The amount of tissue per isolation should be provided. Please add details regarding blocking stage (blocking agent, concentration, time, temperature), blocking agent for primary antibody and details referring to the secondary antibody (catalogue number, dilution factor, diluent, time, temperature).

[3] Please add information on the volume (in milliliters) of cell suspension seeded into each well. For example, in line 125 it is stated that “1 × 106/well MZ-CRC-1 cells were plated on a six-well plate…”.  However, it was not given what volume of cell suspension was introduced into the individual well.

[4] Please specify in the figure legend the concentration of acetate used in experiments presented in Fig. 2B and 2D.

[5] Delete the sentence in lines 67-68.

[6] Please explain the meaning of CPM used in Fig. 4.

[7] Please add in line 40 the following reference: Ratajczak et al., Int J Mol Sci. 2021 Oct 31;22(21):11829.

[8] Supplementary Figure 1 shows the results for line SW1736, although this cell line is not mentioned in the Materials and Methods (the origin, in what medium it was grown, etc.). Please complete this.

[9] The manuscript contains a lot of typos. In many places lack spaces between numbers and units (e.g., in lines 106, 118, 127, 148). The Authors use different milliliter notation (“ml” and “mL” – e.g., in line 308). Other examples of typos:

a.       Line 259: Change “FTC133” and “TPC1” for “FTC-133” and “TPC-1”, respectively.

b.       Lines 227, 346-347: Change “MZ-CRC1” for “MZ-CRC-1”.

c.       Line 259: Change “MZ-CRC” for “MZ-CRC-1”.

d.       Fig. 2C: Change “Ctr” for “Ctrl”.

Author Response

 [1] Please add information on the manufacturer of protease inhibitor cocktail (line 194) used for cell proteins isolation. Moreover, the Authors should specify how long it took to block the membrane before incubation with the primary antibody and at what temperature this was performed. Catalogue numbers, dilution factor and diluent composition for secondary antibodies are needed. Similarly, dilution factor for anti-beta actin antibody (line 204) is not provided. Finally, it is unclear which primary antibody was incubated in the presence of BSA and which in the presence of skimmed milk (line 205).

REPLY1> The manufacturer of the "protease inhibitor cocktail" is Sigma. We have added the information regarding the missing experimental conditions as follows:

MZ-CRC-1 cell lysates were prepared with lysis buffer (50 mM Tris-HCl, 150 mM Sodium chloride, 1% Triton X-100, 1% sodium deoxycholate, 0.1% SDS, pH 7.5, and 2 mM EDTA, 1 mM sodium orthovanadate, 0.1 mM phenylmethylsulfonyl fluoride (PMSF), 0.5% NP-40, protease inhibitor cocktail (Sigma-Aldrich, St. Louis, MO, USA)). The protein concentration was determined by the Bradford assay (Bradford assay kit, Biorad Laboratories, CA, USA) with bovine serum albumin (BSA) as the standard. Equal aliquots of total cell lysates (30 μg) were solubilized in sample buffer and elec-trophoresed on denaturing SDS-polyacrylamide gel (10% and 12% separating gel). The proteins were transferred to polyvinylidene difluoride membranes. The membranes were blocked with 5% nonfat dry milk in TBS containing 0.05% Tween 20 for 1 h at RT and incubated with primary antibodies overnight at 4℃ and then with horseradish peroxidase-conjugated secondary for 2 h at RT. Primary antibodies used CREB (#9197; Cell signaling, MA, USA), phospho-CREB (ser133) (#9198; Cell signaling, MA, USA) and OR51E2 (NBP1-71145; Novus Biologicals, CO, USA). β-Actin (C4) HRP (47778; Santa Cruz Biotechnology, CA, USA) was used as a loading control. The secondary an-tibody used was goat anti-rabbit immunoglobulin-HRP antibody (sc-2004; Santa Cruz Biotechnology). β -Actin was diluted 1:3000 in 5 % nonfat dry milk. CREB, phos-pho-CREB, and OR51E2 were added 1:1000 in 5 % BSA. The secondary antibody was added 1:3000 in 5 % nonfat dry milk. Antigen-antibody complexes were detected with WEST-SAVE Up™ luminol-based ECL reagent (ABfrontier, Seoul, Korea).

[2] Protein isolation from mice tissue: Please indicate the composition of the RIPA buffer and whether a cocktail of protease inhibitors has been added to RIPA buffer. The amount of tissue per isolation should be provided. Please add details regarding blocking stage (blocking agent, concentration, time, temperature), blocking agent for primary antibody and details referring to the secondary antibody (catalogue number, dilution factor, diluent, time, temperature).

    REPLY2: We have added the aforementioned information to the method section.

Protein was extracted from MTC patient specimens. Western blotting was per-formed using established procedures with modifications. Briefly, samples were ho-mogenized in 100 μL RIPA buffer (25 mM Tris-HCl pH 7.6, 150 mM NaCl, 1% NP-40, 1% sodium deox-ycholate, 0.1% SDS with 0.1 mM PMSF and 1×protease inhibitors) using a TissueLyser II (Qiagen, Hilden, Germany). 10 μg of each sample was loaded onto an SDS gel and trans-ferred by electrophoresis to PVDF membrane. Membranes were blocked with 5% non-fat dry milk in TBS containing 0.05% Tween 20 for 1 hour at RT. OR51E2 primary antibody (NBP1-71145; Novus Biologicals CO, USA) was diluted 1:500 in 5% BSA buffer and incu-bated overnight. Blots were washed three times with 0.05% TBS-T buffer, and then incu-bated with HRP-conjugated secondary antibody for 1 h at RT. As the secondary antibody, goat anti-rabbit IgG-HRP antibody (sc-2004; Santa Cruz Bio-technology) was diluted 1:3000 in 5% nonfat dry milk. Western blot results were nor-malized with β -Actin (1:3000, 47778; Santa Cruz Biotechnology). Beta-Actin was diluted 1:3000 in 5% nonfat dry milk and reacted for 1 h at RT. Immunoreactivity was detected with WEST-SAVE UpTM luminol-based ECL reagent (ABfrontier).

[3] Please add information on the volume (in milliliters) of cell suspension seeded into each well. For example, in line 125 it is stated that “1 × 106/well MZ-CRC-1 cells were plated on a six-well plate…”.  However, it was not given what volume of cell suspension was introduced into the individual well.

>REPLY3: For the 6-well plate, 2 mL of cell-contained medium was added to each well. In the 96-well plate, 200 μL was added to each well. In the 60 mm culture dish, 5 mL of medium was added.

2.3. Luciferase assay

The Dual-Glo luciferase assay system (#E2940; Promega, Madison, WI, USA) was used for OR ligand–receptor interaction as previously described [20]. Briefly, 200 μL of MZ-CRC-1 cells (1x104 cells/well) were seeded in white polystyrene 96-well plates (#353296; BD Biosciences) 1 day before transfection. In each well, OR51E2, CRE-luciferase reporter, and Renilla expression vectors were transiently co-transfected into the cells using Lipofectamine 2000. Transfected cells were stimulated at 37 °C for 4 h with various concentrations of acetate diluted in CD293 medium (#11913-019; Thermo Fisher Scientific). The activity of firefly luciferase was normalized against that of Renilla luciferase. Luminescence was measured on a SpectraMax L microplate reader (Molec-ular Devices).

2.4. cAMP assay

cAMP assay was performed according to manufacturer’s instruction (Cyclic AMP XP assay kit, #4339; Cell Signaling Technology, Beverly, MA, USA). Briefly, 5 mL of cells (2 x 106 cells/well) were plated on 60-mm dish, and after overnight adhesion, treated with vehicle or 250 µM acetate and incubated for 1 h. Cells are rinsed with PBS and iced for 10 min after adding 100 μL of 1x lysis buffer. A standard was prepared in 1x lysis buffer and 50 uL HRP-linked cAMP solution and 50ul sample to the cAMP plate was added. After incubation at room temperature (RT) for 3 h on a horizontal or-bital plate shaker, plate contents were discarded and washed 4 times with 1x wash buffer. The buffer in the well was discarded and dried completely. 100 μL TMB sub-strate was added and incubated at RT for 30 min. 100 μL stop solution was added and absorbance at 450 nm was measured.

[4] Please specify in the figure legend the concentration of acetate used in experiments presented in Fig. 2B and 2D.

REPLY> We have added the information that in Figure 2B and 2D, the treatment was performed with 250 µM acetate.

[5] Delete the sentence in lines 67-68.

REPLY>We have deleted as requested. Thank you

[6] Please explain the meaning of CPM used in Fig. 4.

REPLY>We apologize, CPM is counts per minute, it is the radioactive decay units used in C11-acetate decay. We have added this in the manuscript.

[7] Please add in line 40 the following reference: Ratajczak et al., Int J Mol Sci. 2021 Oct 31;22(21):11829. REPLY>We have added as requested. Thank you

[8] Supplementary Figure 1 shows the results for line SW1736, although this cell line is not mentioned in the Materials and Methods (the origin, in what medium it was grown, etc.). Please complete this.

REPLY> We have added information about SW1736 in the manuscript: SW1736 cells were provided by Professor Yoon Woo Koh (Yonsei medical center, Seoul, South Korea). All cell culture reagents were supplied by Hyclone Laboratories, Inc. (Logan, UT, USA).

[9] The manuscript contains a lot of typos. In many places lack spaces between numbers and units (e.g., in lines 106, 118, 127, 148). The Authors use different milliliter notation (“ml” and “mL” – e.g., in line 308). Other examples of typos:

  1. Line 259: Change “FTC133” and “TPC1” for “FTC-133” and “TPC-1”, respectively.
  2. Lines 227, 346-347: Change “MZ-CRC1” for “MZ-CRC-1”.
  3. Line 259: Change “MZ-CRC” for “MZ-CRC-1”.
  4. Fig. 2C: Change “Ctr” for “Ctrl”.

REPLY: We have made the necessary revisions as you pointed out and reviewed for any typos and spacing issues.

Reviewer 2 Report

ID: biomedicines-2322517

Acetate-mediated Odorant Receptor OR51E2 Activation Results in Calcitonin Secretion in Parafollicular C-cells: A Novel Diagnostic Target of Human Medullary Thyroid Cancer. by Lee et al.

To the Authors:

General comment: The authors revised the manuscript according to the comments well; however, some points should be addressed to improve the manuscript.

Specific comment:

1. Description in lines 378-399 needs appropriate references.  Please add them.

Author Response

We have added the references as requested. Thank you.

Reviewer 3 Report

All my commentaries have been satisfactorily replied to. Thank you very much.

Author Response

Thank you for your comment.